# Intramuscular Ganglion Cyst of the Flexor Hallucis Brevis Secondary to Muscle Tear: A Case Report

**DOI:** 10.3390/diagnostics10070484

**Published:** 2020-07-16

**Authors:** Min Cheol Chang, Mathieu Boudier-Revéret, Ming-Yen Hsiao

**Affiliations:** 1Department of Physical Medicine and Rehabilitation, College of Medicine, Yeungnam University, Daegu 42415, Korea; wheel633@ynu.ac.kr; 2Department of Physical Medicine and Rehabilitation, Centre Hospitalier de l’Université de Montréal, Montreal, QC H2X 3E4, Canada; 3Department of Physical Medicine and Rehabilitation, National Taiwan University Hospital, College of Medicine, National Taiwan University, Taipei 100, Taiwan; myferrant@gmail.com

**Keywords:** intramuscular ganglion cyst, ultrasonography, magnetic resonance imaging, flexor hallucis brevis, muscle tear

## Abstract

In the current study, we present a case of an intramuscular ganglion cyst in the flexor hallucis brevis muscle (FHB) that arose secondary to a muscle tear. Through this study, we propose a possible aetiology for the development of intramuscular ganglionic cysts. A 50-year-old woman presented with acute pain and swelling over the right mid-plantar area after prolonged kneeling for scrubbing floors. Ultrasonography examination performed at 5 days after the onset of symptoms revealed a partial tear of the right FHB. Follow-up evaluations were conducted, with magnetic resonance imaging and ultrasonography, at 24 and 54 days after symptom onset. MRI revealed a ganglion cyst in the mid-portion of the FHB without connection to the adjacent joint capsule or tendon sheath. On the ultrasonography examination at 45 days after onset, at the same location where a tear was seen on the initial examination, an anechoic defect in the mid-portion of the FHB was observed, compatible with a ganglion cyst. Given the favourable natural evolution, no aspiration or surgery were performed. The patient was discharged with minimal symptoms. The results suggest that the intramuscular ganglion cyst can develop following a muscle tear.

## 1. Introduction

A ganglion cyst is a common soft tissue mass, usually forming over a joint or tendon, and commonly developing in the wrist on either the dorsal or volar side [1]. It contains a thick and gel-like fluid with a connective tissue capsule, and is known to usually originate from the joint capsule or tendon sheath [1]. It seems that the synovial fluid within the joint or tendon leaks out, is collected, and forms a cyst. In clinical practice, ganglion cysts attached to the joint or tendon are frequently encountered. However, an intramuscular ganglion cyst without connection to the joint or tendon has rarely been reported [2]. In addition, its aetiology has not been clearly elucidated.

In the present study, we describe a case of an intramuscular ganglion cyst in the flexor hallucis brevis muscle (FHB) that arose secondary to a muscle tear. Moreover, through this study, we suggest a possible aetiology for the development of intramuscular ganglion cysts.

## 2. Case Presentation

A 50-year-old woman visited the department of physical medicine and rehabilitation at a university hospital because of acute pain and swelling over the right mid-plantar area for 2 days, which occurred after prolonged kneeling for scrubbing floors. She mentioned that she kneeled with hyperextended toes on her right side for about 3 h, with sudden sharp pain and progressive swelling of the foot developing after work. On physical examination, a focally swollen and mildly erythematous right medial mid-plantar area was noted without ecchymosis (Figure 1).

Palpation revealed heat and tenderness along the right medial mid-plantar area near the first metatarsal shaft, but no palpable mass. There was mild weakness of right big toe flexion. Ultrasonography examination was arranged to evaluate the plantar area. Three days after her first visit to our hospital, ultrasonography (12-MHz linear probe, Toshiba, Tokyo, Japan, Aplio 500) of the right medial mid-plantar area revealed a focal disruption of the fibrillary structure of the FHB with an anechoic defect in the mid-portion of the muscle (Figure 2 and Appendix A). 

The focal defect was confirmed by its compressibility and visibly retracted torn fibres during compression. On the basis of the finding from the ultrasonography examination, a diagnosis of partial tear of the right FHB was made.

At 24 days after the onset of the symptoms, on magnetic resonance imaging (MRI), a smooth, well-circumscribed, unilocular and homogeneously T2-hyperintense lesion (size, 10.0 × 20.9 × 9.1 cm) was found in the mid-portion of the FHB (Figure 3). The MRI finding consisted of a ganglion cyst [2].

In addition, on the follow-up ultrasonography examination 54 days after the onset of the patient’s symptoms, at the same location where a tear was seen on the initial ultrasonography examination, an anechoic defect in the mid-portion of the FHB was observed, filled with a well-defined, unilocular, avascular, anechoic ovoid mass (size, 15.5 × 22.8 × 9.3 cm) with posterior acoustic enhancement, which is the typical appearance of a ganglion cyst (Figure 4 and Appendix A) [3]. The patient’s ganglion cyst was not connected to any tendon or joint. Her pain was significantly reduced to the endurable level; thus, we did not perform an aspiration of the ganglion cyst.

## 3. Discussion

In this study, we presented a rare case of disruption of the flexor hallucis brevis muscle with subsequent development of an intramuscular ganglion cyst, with follow-up evaluation with MRI and ultrasonography.

MRI has been shown to have a high sensitivity and specificity, 94.7% and 94.4%, respectively, in accurately diagnosing ganglion cysts [4]. The data regarding the accuracy of ultrasonography in diagnosing ganglion cysts based on post-surgical pathology analysis is scarce, but a retrospective analysis of 106 patients revealed ultrasonography had a 69% sensitivity and 100% specificity [5]. However, no ganglion cysts were reported on the plantar aspect of the foot in a retrospective analysis of 101 patients surgically treated for foot masses [6]. Therefore, even though some authors have reported US-guided aspiration of ganglion cysts as a potential alternative to surgery [7], clinicians should be very cautious in performing it on lesions in atypical locations.

In most cases, the ganglion cyst can develop from the joint or tendon sheath [1]. However, rarely, a ganglion cyst originates from the muscle without connecting to the adjacent joint capsule or tendon sheath. To the best of our knowledge, seven previous studies reported ganglion cysts confined to a muscle portion with no connection to other adjacent structures [1,8,9,10,11,12,13]. In these studies, intramuscular ganglion cysts were developed in the quadriceps, gastrocnemius, extensor digitorum and biceps brachii. However, no previous study described the primary cause or possible mechanism of the occurrence of the intramuscular ganglion cyst. In our study, with a follow-up examination, we found that the intramuscular ganglion could develop after muscle injury.

The exact mechanism underlying the development of the intramuscular ganglion cyst after muscle tear is unclear. However, on the basis of previous studies, we can suggest a possible mechanism. Some studies reported that the ganglion cyst is thought to arise from the myxoid degeneration of connective tissue that developed from defects of the joint capsule or tendon sheaths [14]. Likewise, we believe that the pieces of torn muscle tissues turn into myxoid fluid and form an intramuscular ganglion cyst in the empty space created after the tearing of the muscle.

In conclusion, we report the case of an intramuscular ganglion cyst in the FHB. We demonstrated that the ganglion cyst could potentially develop after partial muscle tear by follow-up examination with MRI and ultrasonography. In addition, we suggest the myxoid degeneration of fragments of torn muscle as a possible mechanism of the development of the intramuscular ganglion cyst after muscle tear. However, because this study only involves a single case, further studies including a larger number of cases are necessary.

## Figures and Tables

**Figure 1 diagnostics-10-00484-f001:**
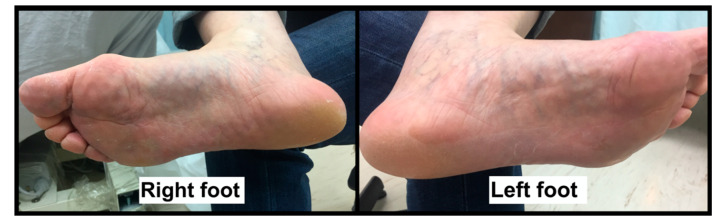
A photograph of the right (symptomatic) and left (asymptomatic) soles of the patient. A mild medial mid-sole bulging can be observed on the right side.

**Figure 2 diagnostics-10-00484-f002:**
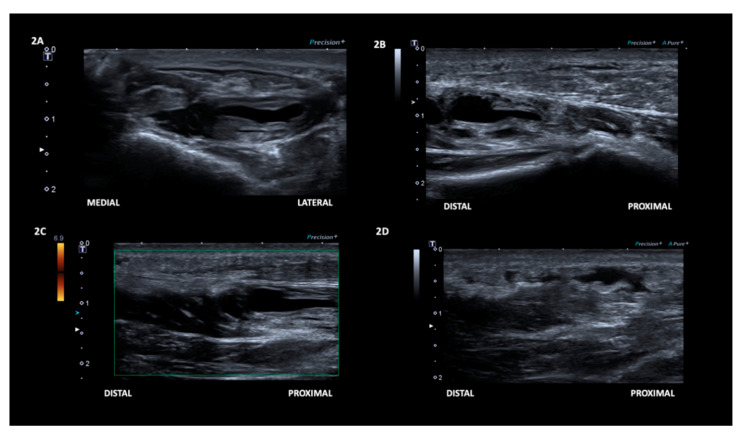
The initial ultrasound examination revealed an area of focal disruption of the fibrillary structure of the mid-portion of the flexor hallucis brevis muscle, shown in transverse axis (**A**), and longitudinal axis (**B**). No vascularity was observed with Power Doppler (**C**). Additionally, localized subcutaneous oedema in close proximity to the zone of muscle tear (**D**).

**Figure 3 diagnostics-10-00484-f003:**
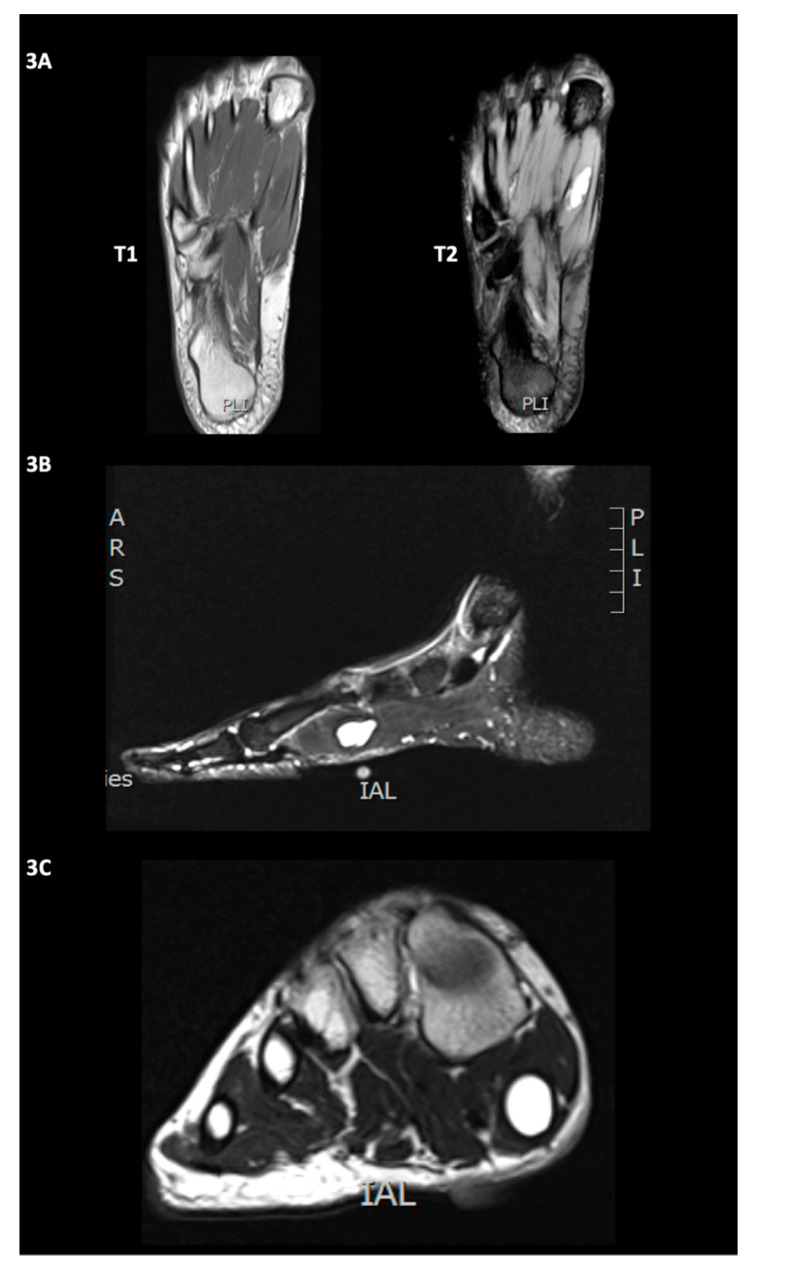
The MRI of the right foot in transverse axis T1 and T2 (**A**), sagittal STIR (TR: 4680, TE: 68, inversion time: 160 ms) (**B**), and coronal T2 (**C**) sequences. It revealed a smooth, well-circumscribed, unilocular and homogeneously T2/STIR-hyperintense lesion (size, 10.0 × 20.9 × 9.1 cm) without noticeable surrounding oedema in the mid-portion of the flexor hallucis brevis muscle, consistent with an intramuscular ganglion cyst.

**Figure 4 diagnostics-10-00484-f004:**
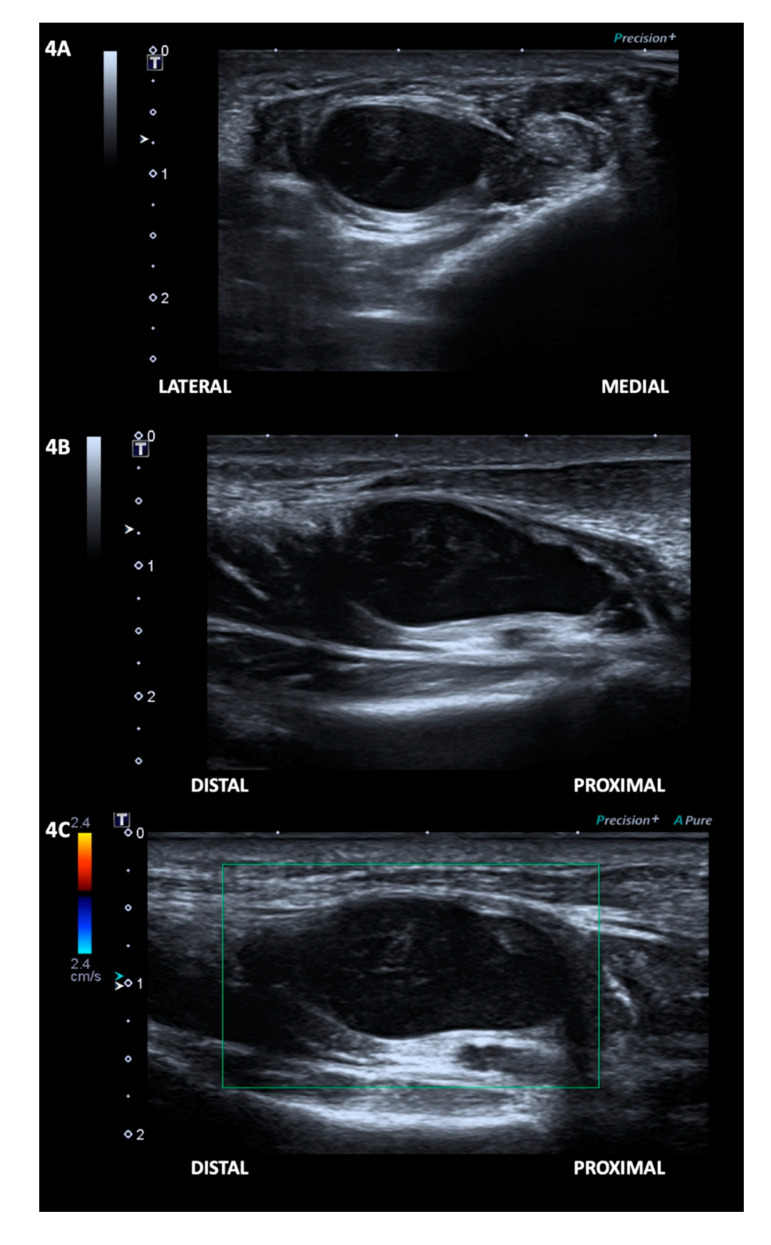
The second ultrasound examination revealed an anechoic defect in the mid-portion of the flexor hallucis brevis muscle, filled with a well-defined, unilocular, anechoic mass with posterior acoustic enhancement, seen in the transverse (**A**) and longitudinal (**B**) axis. Colour Doppler was negative (**C**). The appearance was compatible with a ganglion cyst.

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
