# Peer review of "Intramuscular Ganglion Cyst of the Flexor Hallucis Brevis Secondary to Muscle Tear: A Case Report"

_diagnostics, 2020, doi:10.3390/diagnostics10070484_

Round 1

Reviewer 1 Report

This is an interesting case and it is well presented. My only comments/suggestions are some minor edits to sentence structure (see below):

Line 14-15: Change to: “…we attempt to propose a possible…”

Line 15: Add a space at the end of this sentence.

Line 93: Change to: "...in a retrospective analysis of 101 patients..."

Line 94: Add comma after “Therefore”

Line 95: Change to: "...to surgery, [8] but clinicians..." 

Line 104: Change to ““…can develop..”

Line 106: Change to: “The exact mechanism underlying the development of the…”

Line 127: Change the pronoun to “her

Author Response

This is an interesting case and it is well presented. My only comments/suggestions are some minor edits to sentence structure (see below):

Line 14-15: Change to: “…we attempt to propose a possible…”

Answer: The sentence was modified accordingly. Thank you.

Line 15: Add a space at the end of this sentence.

Answer: It appears that in our original document, a space was already present, but disappeared in the submitted version.

Line 93: Change to: "...in a retrospective analysis of 101 patients..."

Answer: The sentence was modified accordingly. Thank you.

Line 94: Add comma after “Therefore”

Answer: The sentence was modified accordingly. Thank you.

Line 95: Change to: "...to surgery, [8] but clinicians..." 

Answer: The sentence was modified accordingly. Thank you.

Line 104: Change to ““…can develop..”

Answer: The sentence was modified accordingly. Thank you.

Line 106: Change to: “The exact mechanism underlying the development of the…”

Answer: The sentence was modified accordingly. Thank you.

Line 127: Change the pronoun to “her”

Answer: We are uncertain if the reviewer wants us the change “the patient” to “her”, but we left the sentence as it was.

Reviewer 2 Report

The present case report describes how an intramuscular ganglion cyst in the flexor hallucis brevis muscle could arise secondary to a muscle tear, in a clear, planar and exhaustive language.

Author Response

The present case report describes how an intramuscular ganglion cyst in the flexor hallucis brevis muscle could arise secondary to a muscle tear, in a clear, planar and exhaustive language.

Thank you for your comments

Reviewer 3 Report

The introduction section needs to be detailed, as the authors did not explain the importance and the prevalence of the ganglion cyst.

Fig 3; 3B and 3C may be explained better

Results: The authors used the terminology ' rupture
' in discussion and 'disruption' in results. The authors may use a consistent term throughout the manuscript.

The authors sted that ' 7 previous studies reported ganglion cysts
', which may be detailed.

How did the authors concluded that ' intramuscular ganglion can be developed after muscle injury'?

The discussion should be detailed, in the light of previous studies.

Author Response

Comments and Suggestions for Authors

The introduction section needs to be detailed, as the authors did not explain the importance and the prevalence of the ganglion cyst.

Answer: We added information about the prevalence of ganglion cyst in the foot & ankle as well as on its importance.

Fig 3; 3B and 3C may be explained better

Answer: The caption of Fig 3 was modified accordingly. Thank you.

Results: The authors used the terminology ' rupture' in discussion and 'disruption' in results. The authors may use a consistent term throughout the manuscript.

Answer: We used disruption throughout the text following the reviewer’s comment.

The authors stated that ' 7 previous studies reported ganglion cysts', which may be detailed.

Answer: We detailed the findings of these 7 studies (lines 129-134).

How did the authors concluded that ' intramuscular ganglion can be developed after muscle injury'?

Answer: This is what is unique about this case: given the initial images and clinic of FHB muscle disruption, that evolved to an MRI-confirmed and US-compatible intramuscular ganglion cyst, we concluded that “intramuscular ganglion can be developed after muscle injury”. Even though this mechanism has been alluded to in other studies, none had imaging proof that a previous muscle tear had occurred. We hope this explanation is satisfactory to the reviewer.

The discussion should be detailed, in the light of previous studies.

Answer: As mentioned above, we detailed the findings other studies reporting intramuscular ganglion cysts in the discussion (lines 129-134).

Round 2

Reviewer 3 Report

The authors may list out the risk factors and also the causes of the ganglion cysts in the introduction section

Author Response

Comment: The authors may list out the risk factors and also the causes of the ganglion cysts in the introduction section

Answer: The aetiology of ganglion cysts remains unclear, as stated in our article. Even for cysts of articular or tendon sheath origin, many theories exist and there is no definitive answer on the origin of the fluid and the ganglion. In their review article, Gude & Morelli (Gude, W., & Morelli, V. (2008). Ganglion cysts of the wrist: pathophysiology, clinical picture, and management. Current reviews in musculoskeletal medicine, 1(3-4), 205-211.) mention that "Theories on cyst genesis have been difficult to prove and most are unable to account for all of the known features of the ganglion cyst." Even though joint degenerative changes and tendon/tendon sheath overload have often been thought to be risk factors to develop ganglion cyst, they are not prerequisites in all cases and therefore might be only "associative" factors.

Also, the risk factors often mentioned on health institution websites geared toward the general population (e.g. https://www.mayoclinic.org/diseases-conditions/ganglion-cyst/symptoms-causes/syc-20351156 ; https://www.uofmhealth.org/conditions-treatments/cmc/hand-elbow-wrist/ganglion-cysts) of age (20-40 age group) and sex (female) are not mentioned in most review articles on the topic.

Given that our case pertains more specifically to intramuscular ganglion cyst, which is a rarer entity with unclear pathophysiology (on which we elaborate in the discussion), we think it is best not to mention "weak"/uncertain risk factors or associations in the introduction.